# Relationship between trust and patient involvement in medical decision-making: A cross-sectional study

**Irina Pokhilenko**[1]*, **Thamar E. M. van Esch**[2], **Anne E. M. Brabers**[2], **Judith D. de Jong**[1,2]

**1** Faculty of Health, Department of Health Services Research, Care and Public Health Research Institute, Medicine and Life Sciences, Maastricht University, Maastricht, The Netherlands, **2** Nivel, the Netherlands Institute for Health Services Research, Utrecht, The Netherlands

* i.pokhilenko@maastrichtuniversity.nl

## Abstract

### Introduction

Patients vary in their preferences regarding involvement in medical decision-making. Current research does not provide complete explanation for this observed variation. Patient involvement in medical decision-making has been found to be influenced by various mechanisms, one of which could be patients' trust in physicians. The aim of this study was to examine whether trust in physicians fosters or impairs patient involvement in medical decision-making. This study also aimed to determine to what extent the relationship between trust and preferences regarding decision-making roles was influenced by the sociodemographic characteristics of the patients. We hypothesised that trust can both foster and impair patient involvement in medical decision-making.

### Materials and methods

A survey was sent out to members of the Nivel Dutch Health Care Consumer Panel in February 2016 (response rate = 47%, N = 703). The Wake Forest Physician Trust Scale was used to measure trust. Patient involvement was measured using two items based on the study published by Flynn and colleagues in 2006. Multiple regression analysis was used to analyse the relationship between trust and patient involvement.

### Results

We found a negative relationship between trust and patient involvement in medical decision-making in men. Women with high trust reported to be more involved in medical decision-making compared to men with high trust.

### Conclusion

The results suggest that trust impairs involvement in medical decision-making for men but not for women. Further research could provide a more comprehensive explanation of the

**Data Availability Statement:** The minimal anonymized data set necessary to replicate our study findings are uploaded to DANS EASY repository (https://doi.org/10.17026/dans-x3y-

5f6k) and will be made available upon request. Additionally, the same minimal anonymized data set is available upon request from prof Judith D. de Jong, PhD (j.dejong@nivel.nl), project leader of the Dutch Health Care Consumer Panel, or from the secretary of this panel (consumentenpanel@nivel. nl). The Dutch Health Care Consumer Panel has a program committee, which supervises processing the data of the Dutch Health Care Consumer Panel and decides about the use of the data. This program committee consists of representatives of the Dutch Ministry of Health, Welfare and Sport, the Health Care Inspectorate, Zorgverzekeraars Nederland (Association of Health Care Insurers in the Netherlands), the National Health Care Institute, the Federation of Patients and Consumer Organisations in the Netherlands, the Dutch Healthcare Authority and the Dutch Consumers Association. All research conducted within the Consumer Panel has to be approved by this program committee. The committee assesses whether a specific research fits within the aim of the Consumer Panel, that is strengthen the position of the health care user.

**Funding:** The authors received no specific funding for this work.

**Competing interests:** The authors have declared that no competing interests exist.

variation in patient preferences regarding involvement in medical decision-making to further elucidate which underlying mechanisms could enhance patient participation.

## Introduction

The last few decades have witnessed a major shift from paternalism to shared decision-making (SDM) in the patient-physician relationship [1], which led to an increasing emphasis on patients' active participation in medical decision-making (MDM). SDM represents one of the approaches to MDM [2]. In contrast to the paternalistic model, where physicians impose their decisions on the patients, SDM is "the process through which clinicians and patients share information with each other and work toward decisions about treatment chosen from medically reasonable options that are aligned with the patients' values, goals, and preferences" [3]. The idea of SDM has emerged from the principles of patient-centred care, which is defined as "care that is respectful of and responsive to individual patient preferences, needs, and values" and that ensures "that patient values guide all clinical decisions" [4]. Therefore, SDM is an essential attribute of patient centred care and good quality of care as it implies that patient preferences are taken into account [5].

Previous studies indicate that most patients, especially younger, higher educated, and female patients prefer a shared approach to MDM [6–9]. Additionally, patients' experience of illness and medical care, their diagnosis and health status, the type of decision that needs to be made, the amount of knowledge patients have acquired about their condition, their attitude towards SDM, and the patient-physician relationship play a role in forming patient preferences about involvement in MDM [6]. It can be concluded that patients vary in their preferences for involvement in MDM and that these preferences are developed as a consequence of a highly complex process under the influence of multiple factors mentioned above [6]. A number of mechanisms including critical high health literacy, availability of medical informational support, and non-conservative social norms were found to positively influence patient involvement in MDM [10–12]. However, the studied mechanisms account for only a part of the observed variation in patient preferences for involvement in MDM [10–12]. To provide a more comprehensive explanation, other mechanisms that might influence patient involvement need to be studied as well. One of such mechanisms might be trust.

This study examines trust in physicians as a mechanism that might further explain the variation in patient preferences regarding participation in MDM. In this study, we focus on the interpersonal trust of patients in their physician, which can be defined as "the optimistic acceptance of a vulnerable situation in which the patient believes that the physician will care for the patients' interests" [13]. On the one hand, trust in physicians is considered to be an important aspect of a therapeutic relationship, a key predictor of patient involvement in MDM [7], and an important source of support, especially for critically ill patients, helping them deal with uncertainty and fear [14]. On the other hand, trust can also prevent patients from being truly autonomous and taking the responsibility for their own health, thereby hindering an optimal patient-physician relationship [15]. Therefore, patients' trust in physicians can act both as a barrier and a facilitator of patient involvement in MDM [16–21].

### Relationship between trust and patient involvement in medical decision-making

Although earlier studies found that higher trust in physicians can both foster and impair involvement in MDM [16,20,22], more insight into the mechanisms of this relationship is

needed. Since trust in physicians is influenced by patients' sociodemographic characteristics [21,23,24], investigating how sociodemographic characteristics affect the relationship between trust and patient involvement in MDM might provide an explanation of why for some patients high trust acts as a facilitator of their involvement in MDM, while for others it can be a barrier. Furthermore, previous studies have been performed either among specific patient groups including patients of breast cancer, prostate cancer and fracture outpatient clinics [21], African-American patients with diabetes [16,17], patients with cardiovascular disease [20] or among a group of breast cancer specialists [25]. Trachtenberg and colleagues (2005) examined a random national sample in the US; however, the sample was not fully representative of the general American population [7]. Since the samples of earlier studies examine various specific groups, generalizability of the findings is in question. Therefore, this study was performed among a representative sample of the general population in the Netherlands, which is likely to increase the generalizability of the findings.

## Relationship between trust and sociodemographic characteristics

Several studies also examined the relationship between sociodemographic characteristics and trust. These studies established some correlation between the level of trust and age [23,24], gender [21,23], and educational level [21,23]. Nevertheless, according to Hall and colleagues (2001), among others, demographic characteristics, with the exception of age, are not strong predictors of trust [13]. Because previous studies produced conflicting results, this study examined the relationship between the level of trust and age, gender, and educational level in the sample of a general population in the attempt to explain the bidirectionality of the relationship between trust and patient involvement in MDM.

## Aims of the study

The aims of this study were to examine the relationship between patients' trust in their physician and their involvement in MDM and to determine the extent to which this relationship was influenced by sociodemographic characteristics of the patients. This study was performed among a sample of the general population aged 18 years and older in the Netherlands and focussed on the self-reported patient involvement in MDM.

## Hypothesis and conceptual model of the study

Based on the aims of the study and the reviewed literature concerning the relationship between trust and patient involvement as well as predictors of trust, several hypotheses were developed.

It is hypothesized that higher level of trust can both foster and impair patient involvement in MDM [16,20,21]. A trusting patient-physician relationship motivates patients to engage in MDM, because they feel comfortable to speak up and ask questions [17]. On the other hand, high level of trust might also encourage passivity [7,21]. Patients with high level of trust are more likely to assume a passive role [21]; they believe that their physician acts in their best interest and, thus, do not deem their involvement necessary [22]. Therefore, the first hypothesis is:

**H1: High trust can both foster and impair patient involvement in medical decision-making**.

Older patients tend to have higher levels of trust in their physician in comparison to younger patients [23,24]. This is likely due to the generational effect, which means that the elderly have more respect towards physicians compared to younger generations [13]. The association between older age and higher trust might also be related to the fact that older patients are more likely to have more contacts and a generally longer relationship with their physician and,

therefore, develop higher level of trust [13]. On the other hand, younger people use less healthcare [26] and are less likely to develop trust in their physician during infrequent visits. Older patients also tend to have a lower level of health literacy [27,28], which makes it more difficult for them to obtain, process and understand basic health information and services needed to make appropriate health decisions [29]. Additionally, elderly find SDM less important [30] and prefer a more passive role in MDM [6]. It can be assumed that low health literacy prevents older people from engaging in MDM and higher trust in their physician might help them cope with vulnerability. Based on this, we hypothesize that:

**H2: Older people with higher trust in their physician are less involved in medical decision-making, while younger people with higher trust are more involved in medical decision-making**.

Women are presumed to have more trust in their physician compared to men [31,32]. Patients who have a longer relationship with their physicians are more likely to trust their physicians [33]. Women are found to use more healthcare services compared to men [34–36], and are, therefore, more likely to develop a longer relationship with their physician, which might explain why women tend to exhibit higher levels of trust compared to men. Besides, women generally prefer a shared approach to MDM [6–9,37]. In addition, women are generally more risk averse than men in the context of physical health and safety [38–41]. It can be hypothesized that women attempt to control the risk by actively participating in MDM. Based on this, it is hypothesized that:

**H3: Women with higher trust in their physician are more involved in medical decision-making, while men with higher trust are less involved in medical-decision making**.

Less educated people are presumed to have higher trust in their physician as opposed to highly educated people [7,21,23,42], which might be attributed to differences in health literacy [27,28]. In addition, less educated people find SDM less important [30] and prefer a more passive role in MDM [6]. It can be assumed that, similarly to the elderly, lower health literacy prevents less educated people from engaging in MDM and that trust helps them cope with vulnerability. Therefore, we hypothesize that:

**H4: Lower educated people with higher trust in their physician are less involved in medical decision-making, while higher educated people with higher trust are more involved in medical decision-making**.

## Methods

### Setting

The data for this study were collected in the Dutch Health Care Consumer Panel, which gathers information about opinions and experiences of the general population in relation to healthcare. Themes of the questionnaires include, among other, trust, solidarity, health insurance, opinions on healthcare insurers and care providers, and eHealth. The Dutch Health Care Consumer Panel is an access panel, which consists of a large number of members that agreed to respond to questionnaires on a regular basis. Certain background characteristics such as age, gender, and self-reported health status of the panel members are known. The panel allows to draw samples for individual studies that are representative with respect to gender and age of the general Dutch population age 18 years and older. The members are recruited in two manners. First, addresses of potential participants are bought from an address supplier. Second, new members are recruited through general practitioner practices that participate in the Primary Care Database programme. People cannot sign up to become a member of the panel on their own initiative. Surveys and questionnaires are being distributed approximately eight times per year. Every member receives a questionnaire about three times a year and is free to

quit the panel at any time. Members are able to choose whether they prefer to fill out paper-based or online questionnaires. There are no costs associated with the membership. At the moment of data collection (February 2016), the Dutch Health Care Consumer Panel consisted of approximately 12,000 members of 18 years or older [43]. As of 2021, the number of panel members is approximately 11,500. According to the Dutch legislation, neither obtaining informed consent nor approval by a medical ethics committee is obligatory for conducting research through the panel [44]. Data are analysed anonymously, and processed according to the privacy policy of the Dutch Healthcare Consumer Panel, which complies with the General Data Protection Regulation (GDPR). A privacy regulation is available for the Consumer Panel [45]. The questionnaires used in this study were anonymous in nature. Besides, the researchers who analysed the data did not have access to any identifiable information of the participants (e.g. name and address).

For this study, a questionnaire was sent to a sample of 1,500 panel members in February 2016. The sample was randomly selected and was representative of the Dutch adult population for age and sex. The survey inquired about various topics. For this study, only two sections of the survey related to patient involvement in MDM and trust in their physician were assessed. Two electronic reminders (after one and two weeks) and one postal reminder (after two weeks) were sent out to the panel members who did not return the questionnaires. The closing date for the questionnaire was four weeks after the initial distribution. The respondents were not obliged to answer the questionnaire. 703 panel members returned the questionnaire (response rate of 47%).

## Measurements

**Trust.** Trust was measured using the Dutch version of the "Wake Forest Physician Trust Scale" (WF-D) [46], which consists of one question subdivided in ten items (Table 1). The respondents were asked to indicate to what extent they trust their physician on a 5-point Likert

**Table 1. Measurement of trust: Wake Forest Physician Trust Scale.**

| Questions | Answer categories |
|---|---|
| Indicate the extent to which you agree with the following statements: | Completely disagree (score 1), disagree (score 2), neither agree or disagree (score 3), agree (score 4), completely agree (score 5) |
| **a**. Your physician will make every effort to ensure that you receive the care you need. | |
| **b**. Sometimes your physician sets his/her own interests above your medical interest (reverse coded) | |
| **c**. Your physician's medical skills are not as good as they should be (reverse coded) | |
| **d**. Your physician is extremely careful and accurate. | |
| **e**. You have every confidence in your physician's decision about which medical treatments are best for you. | |
| **f**. You physician informs you in all fairness about the different treatments available for your condition. | |
| **g**. Your physician thinks only about what is best for you. | |
| **h**. Sometimes your physician does not pay full attention to what you are trying to tell him/her (reverse coded) | |
| **i**. You do not worry about putting your life in the hands of your physician. | |
| **j**. All in all, you trust your physician completely. | |

**Table 2. Measurement of patient involvement in medical decision-making (based on Flynn et al. 2006 [9]).**

| Questions | Answer categories |
|---|---|
| **1**. How often do you think that: | Never (score 1), sometimes (score 2), often (score 3), always (score 4) |
| **a.** You let your physician decide what is the best for your health? (reverse coded) | |
| **b.** The most important medical decisions will be taken by your physician and not by yourself? (reverse coded) | |

scale ("completely disagree" = score 1; "completely agree" = score 5). Items b, c, and h contained negative statements and were, therefore, reverse coded. Only respondents that filled out all questions were included; based on this 70 participants were excluded. The internal consistency of the scale is high as indicated by the Chronbach's alpha (0.9). The total score for each participant was calculated by averaging the responses. The scores ranged from 1 to 5 with higher scores indicating higher trust.

**Patient involvement in medical decision-making.** Two questions based on the items developed by Flynn et al [9] were used to measure patient involvement in MDM (Table 2). The respondents were asked to indicate to what extent they were involved in MDM. Both questions contained negative statements and were, therefore, reverse coded. For each respondent that filled out all questions (n = 40 excluded), the mean score was calculated. The internal consistency given by Cronbach's alpha is 0.68, which is considered acceptable [47]. The scores ranged from 1 to 4; the higher the score, the more actively the respondent stated to be involved in MDM.

**Sociodemographic characteristics.** The following sociodemographic characteristics of the respondents were assessed: age (1 = young, below 40; 2 = middle, between 40 and 64; 3 = elderly, 65 and above); gender (1 = male, 2 = female); educational level (1 = low, 2 = middle, 3 = high). Low educational level referred to none, primary school or prevocational education. Middle educational level referred to secondary or vocational education. High educational level referred to professional higher education or university.

## Statistical analysis

First, descriptive statistics were populated to describe the characteristics of the sample. Second, the relationship between trust and sociodemographic characteristics was tested in a multiple regression model with trust as a dependent variable and the sociodemographic characteristics as independent variables. To analyse the relationship between trust and patient involvement a Pearson correlation was calculated. To examine whether the relationship between trust and patient involvement differs between groups of respondents, a second multiple regression analysis was performed. In the second analysis, patient involvement was included as the dependent variable, trust and sociodemographic characteristic were included as the independent variables. The interactions between trust and sociodemographic characteristics were also included in the second analysis. To carry out statistical analyses, STATA, version 14.0, was used. P<0,05 was considered to be statistically significant.

## Results

### Descriptive statistics

Table 3 describes the characteristics of the respondents. Half (50%) of the respondents were female; the mean age of the respondents was 56 years (range 21 to 91). The majority of the respondents had a middle (54%) or high (31%) level of education. The mean score for patient

**Table 3. Descriptive statistics of the respondents.**

| Characteristic | N | Category | N | %/mean (SD)* |
|---|---|---|---|---|
| **Gender** | 703 | Male | 351 | 49.9 |
| | | Female | 352 | 50.1 |
| **Age** | 703 | | | 56 (15.7)* |
| | | Male | 351 | 56.4 (15.4)* |
| | | Female | 352 | 55.7 (15.9)* |
| **Educational level** | 685 | Low (non, primary school or pre-vocational education) | 103 | 15 |
| | | Middle (secondary or vocational education) | 368 | 53.7 |
| | | High (professional higher education or university) | 214 | 31.2 |
| **Questionnaire** | 703 | Post | 354 | 50.4 |
| | | Internet | 349 | 49.6 |
| **Trust** | 633 | | | 4.03 (.71)* |
| **Involvement in medical decision-making** | 663 | | | 2.43 (.71)* |

involvement was 2.43 (SD = 0.71) on a scale from 1 (no involvement) to 4 (active involvement). Most respondents had a score around 2.5, which means that on average the respondents tended to take medical decisions together with their physician. The mean score for trust was 4.03 (SD = 0.71) on a scale from 1 (low trust) to 5 (high trust). Most respondents (93%) had a score of 3 or higher. The mean score of 4.03 among the respondents indicates relatively high level of trust in their physician.

## Test of the hypotheses

First, the general relationship between trust and patient involvement in MDM was analysed. The analysis indicated that the relationship between trust and patient involvement without controlling for sociodemographic characteristics was negative (Pearson correlation -0.204; p<0.001), which means that higher level of trust was associated with lower patient involvement in MDM. Second, we looked at the relationship between trust and sociodemographic characteristics (Table 4). We found that middle aged (coef. = 0.223; p = 0.003) and elderly people (coef. = 0.357; p = 0.000) had significantly higher trust in their physician compared to younger people. No relationship between trust and gender and trust and educational level was found.

Third, we tested the relationship between trust and patient involvement controlled for the sociodemographic characteristics (Table 5). Interactions between age and trust, educational

**Table 4. Relationship between trust and sociodemographic characteristics, results from the regression analysis.**

| Trust | | | |
|---|---|---|---|
| | Category | Coefficient | P value |
| **Gender** | Male | Reference category | |
| | Female | -0.033 | 0.565 |
| **Age** | Young | Reference category | |
| | Middle aged | **0.223** | **0.003** |
| | Elderly | **0.357** | **0.000** |
| **Educational level** | Low | Reference category | |
| | Middle | 0.090 | 0.280 |
| | High | 0.062 | 0.501 |
| **Constant** | 3.774 | | |

**Table 5. Relationship between patient involvement in MDM and trust, including interactions with sociodemographic characteristics, results from regression analysis.**

| Patient involvement in MDM | | | |
|---|---|---|---|
| | Category | Coefficient | P value |
| **Trust** | | **-0.315** | **0.031** |
| **Age** | Young | Reference category | |
| | Middle | -0.002 | 0.996 |
| | Elderly | -0.221 | 0.679 |
| **Educational level** | Low | Reference category | |
| | Middle | 0.027 | 0.953 |
| | High | 0.542 | 0.280 |
| **Gender** | Male | Reference category | |
| | Female | -0.612 | 0.057 |
| **Interaction between age and trust** | Young | Reference category | |
| | Middle | 0.010 | 0.928 |
| | Elderly | 0.027 | 0.835 |
| **Interaction between educational level and trust** | Low | Reference category | |
| | Middle | 0.055 | 0.618 |
| | High | -0.046 | 0.706 |
| **Interaction between gender and trust** | Male | Reference category | |
| | **Female** | **0.191** | **0.015** |
| **Constant** | 3.393 | | |

level and trust, and gender and trust were also included in the model. The relationship between trust and patient involvement in MDM in this model was negative (Pearson correlation -0.315, p = 0.031). In this model, no significant associations between the sociodemographic characteristics and patient involvement in MDM were found. We found a positive interaction between trust and gender among women compared to men (Table 5, coef. = 0.191; p = 0.015). Fig 1 illustrates the relationship between trust and patient involvement in men and women. Women with higher trust were found to be more involved in MDM compared to men with higher trust. Furthermore, the line that illustrates the relationship between trust and patient involvement in MDM in women is significantly flatter compared to the line illustrating the same relationship in men. Among men higher trust was associated with less involvement in MDM, while the relationship between trust and involvement in MDM among women was not significant. There were no significant interactions between age and trust and educational level and trust.

Based on our findings, we could analyze the hypotheses of the study.

**H1: High trust can both foster and impair patient involvement in medical decision-making**.

The relationship between trust and patient involvement in MDM was shown to be significantly negative for men, while for women there was only a weak negative, non-significant, correlation (Fig 1). This means that trust impairs patient involvement in MDM among men, but not among women. Therefore, the first hypothesis was partially confirmed.

**H2: Older people with higher trust in their physician are less involved in medical decision-making, while younger people with higher trust are more involved in medical decision-making**.

No effect of age on the relationship between trust and patient involvement in MDM was observed. Therefore, the second hypothesis was not accepted.

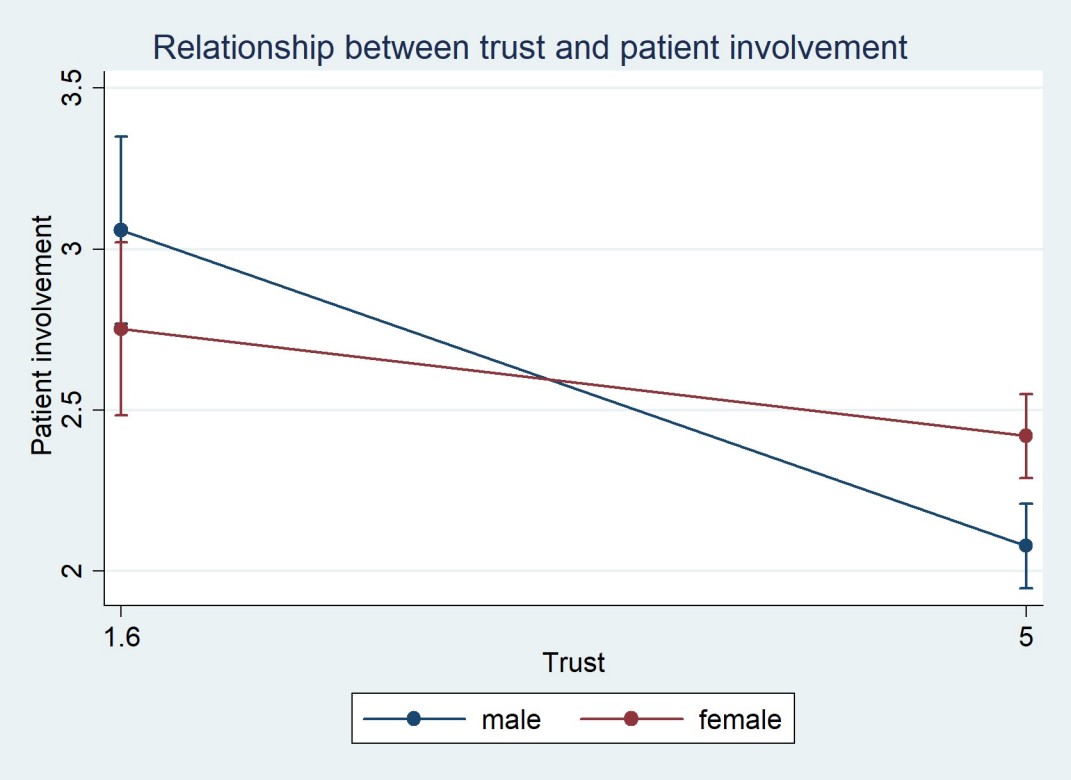

**Fig 1. Relationship between trust and involvement in medical decision-making in men and women.**

**H3: Women with higher trust in their physician are more involved in medical decision-making, while men with higher trust are less involved in medical-decision making**.

Gender was found to have an effect on the relationship between trust and patient involvement in MDM. This relationship was significantly less negative among women compared to men. Fig 1 shows that women with higher trust were found to be more involved in MDM compared to men with higher trust. Therefore, the third hypothesis was confirmed.

**H4: Lower educated people with higher trust in their physician are less involved in medical decision-making, while higher educated people with higher trust are more involved in medical decision-making**.

Educational level was found to have no effect on the relationship between trust and patient involvement in MDM. Therefore, the fourth hypothesis was not accepted.

## Discussion

The aims of this study were to examine the relationship between trust in physicians and patient involvement in MDM and to determine how this relationship is influenced by sociodemographic characteristics of the patients. The relationship between trust and patient involvement in MDM was found to be generally negative. This means that people with higher trust in physicians prefer to be less involved in MDM and people with lower trust prefer to be more involved in MDM. However, we found that for women this relationship was different than for men. Women with higher trust preferred to be more involved in medical decision-making compared to men with higher trust.

Although higher trust was associated with less involvement in MDM among men, it is crucial to interpret the results with the consideration of the respondents' characteristics. The level

of trust among the respondents in our study was generally high (Table 3), which means that in the group of people with lower trust the actual level of trust was still relatively high based on the scale. This could be attributed to the generally high level of trust in physicians among the Dutch population [48]. Additionally, on average the respondents preferred a shared approach to MDM (Table 3). Based on this, we expect that the differences between the different sub-groups were not large enough to capture significant relationships between the variables. This has to be taken into account in terms of transferability of the findings, as the results of this study might not be applicable to other settings where the differences between the subgroups are larger.

The relationship between trust and involvement in MDM among women was significantly more positive compared to men, meaning that women with higher trust were more involved in MDM compared to men with higher trust. Furthermore, we found that the relationship between trust and involvement in MDM was weaker for women than for men. This might indicate that other mechanisms explaining the variability of patient preferences regarding their involvement in MDM such as health literacy, availability of medical informational sup-port, or social norms are more important for women. Additionally, we found a strong positive relationship between age and trust; older respondents reported to have more trust in physi-cians compared to younger respondents. However, no direct association between age and patient involvement in MDM was found in a model including trust. Based on the established relationship between trust and involvement in MDM and the relationship between age and trust, we can conclude that age might have an indirect effect on patient involvement in MDM through trust. The educational level of the respondents was not associated with trust, with patient involvement in MDM, and with the relationship between trust and patient involve-ment in MDM. The fact that we did not find a relationship between educational level and involvement in MDM among the respondents is not in line with the literature [8,9]. This could, however, potentially be attributed to the overrepresentation of the respondents with middle and high educational level in this study and could be explored further in a more repre-sentative sample.

Our findings contribute to the understanding of the relationship between trust and patient involvement in MDM in the general population in particular by providing a deeper insight into how this relationship is influenced by sociodemographic characteristics of the respon-dents. In this study the survey was sent out to a large sample size (N = 1,500) that was represen-tative of the general Dutch population. Although the response rate of 47% was relatively low, the number of respondents (n = 703) was large enough to ensure the generalizability of the results to the larger population in the Netherlands. However, it is important to note that the respondents of this study were not entirely representative of the Dutch general population. Younger and less educated people were underrepresented, while middle aged, elderly and higher educated were overrepresented in the group compared to the general Dutch population [43].

We used a validated instrument (WF-D) to measure the level of interpersonal trust of the respondents in their physician. It is important to note that WF-D was not fully balanced in terms of the nature of the statements. It contained seven positive statements and three negative statements, which might have led to acquiescence bias. However, WF-D has been validated [46] and frequently used for trust research [49,50]. Furthermore, this study focussed on the interpersonal trust of patients in their physician. Although Trachtenberg and colleagues (2005) found that general trust in the medical profession is significantly stronger associated with patient involvement compared to the interpersonal trust in a specific physician [7], it was not specified how trust was measured in that study. Additionally, scales to assess trust in a spe-cific physician are more advanced compared to the scales assessing trust in medical institutions

or in the medical profession in general [13]. To assess patient involvement in MDM, adapted questions by Flynn et al. were used. Although these questions have not been validated, they were used previously in multiple studies [10–12]. Additionally, patient involvement was measured through patients' self-assessment instead of observing actual behaviour and it is not clear whether the respondents reported their actual or perceived participation in MDM, which could have affected the validity of our results. Both instruments used in this study did not refer to a particular physician (e.g. respondent's general practitioner). This might have affected the accuracy of the responses in case some of the respondents had more than one treating physician. Finally, to collect the data for this study a cross-sectional survey was conducted. A limitation of the study design is that it only allowed us to test associations, and not causal relationships between the variables. Therefore, we cannot conclude whether higher trust leads to more or less active patient involvement or vice versa.

Our hypotheses were based on the existing literature. For further research, it would be interesting to test the generalizability of our hypotheses in a general population sample with a larger range of trust. An example of such a setting is the US, where the average level of trust in healthcare providers is significantly lower compared to the Netherlands [51]. The hypotheses of this study could also be tested using a mixed methods approach, which would entail observing actual involvement in MDM among the participants in addition to the measurement of the perceived involvement. This would allow the researchers to gain insight into the differences between perceived and actual involvement in MDM and improve the validity of the results. Employing a qualitative study design could also be interesting for investigating the reasons for why women with high trust in their physician are more involved in MDM compared to men with high trust. This could help design strategies for facilitating the involvement of men with high trust in MDM. Furthermore, trust is only one of the mechanisms that explains the variability in patients' preferences for involvement in MDM and we found it to be less important for women. Other mechanisms including high critical health literacy, availability of medical informational support, and non-conservative social norms [10–12] were found to positively influence participation in MDM. To provide a more comprehensive explanation of variability in patient preferences regarding involvement in MDM, future research could focus on exploring the relationship between these mechanisms and patient involvement in one sample. Finally, further research could use longitudinal data to gain insight into causal relationships between the variables.

Optimal patient-physician relationship requires trust as well as patient's active participation in decision-making. Therefore, clinical practice requires an approach that can foster both trust and patient involvement in MDM. In our study, we observed that women with higher trust are more actively involved in MDM. This poses a question whether there is a possibility to design strategies that could potentially reverse the negative association between trust and participation in MDM among men. The knowledge developed by this study could help develop such strategies and interventions. This would also lead to multiple health benefits for the patients, since SDM is generally associated with greater adherence to treatment, patient satisfaction and superior health outcomes [52,53].

## Conclusion

The results of this study suggest that trust is negatively associated with patient involvement in MDM among men. Women with high trust were more involved in MDM compared to men with high trust. Further research is recommended to provide a more comprehensive explanation of the variability in patient preferences regarding involvement in MDM by simultaneously looking at various mechanisms that were previously found to affect patient involvement in

MDM. This knowledge would be important for developing strategies to enhance involvement in MDM among various patient groups.

## Author Contributions

**Conceptualization:** Irina Pokhilenko, Thamar E. M. van Esch, Anne E. M. Brabers, Judith D. de Jong.

**Data curation:** Judith D. de Jong.

**Formal analysis:** Irina Pokhilenko, Thamar E. M. van Esch.

**Investigation:** Irina Pokhilenko, Thamar E. M. van Esch, Anne E. M. Brabers, Judith D. de Jong.

**Methodology:** Irina Pokhilenko, Thamar E. M. van Esch, Anne E. M. Brabers, Judith D. de Jong.

**Project administration:** Judith D. de Jong.

**Supervision:** Thamar E. M. van Esch, Anne E. M. Brabers, Judith D. de Jong.

**Writing – original draft:** Irina Pokhilenko.

**Writing – review & editing:** Irina Pokhilenko, Thamar E. M. van Esch, Anne E. M. Brabers, Judith D. de Jong.

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
