## [Decision Letter · Decision Letter 0]

30 May 2021

PONE-D-21-10608

Relationship between Trust and Patient Involvement in Medical Decision-Making: A Cross-Sectional Study

PLOS ONE

Dear Dr. Pokhilenko,

Thank you for submitting your manuscript to PLOS ONE. After careful consideration, we feel that it has merit but does not fully meet PLOS ONE’s publication criteria as it currently stands. Therefore, we invite you to submit a revised version of the manuscript that addresses the points raised during the review process.

We look forward to receiving your revised manuscript.

Kind regards,

Prof. Ritesh G. Menezes, M.B.B.S., M.D., Diplomate N.B.

Academic Editor

PLOS ONE

Journal Requirements:

Additional Academic Editor Comments:

• Abstract-Line 33: Mention ‘’Flynn et al. (2006).’’ as a part of a complete sentence (avoid the bracket and accordingly edit the sentence.

• Introduction-Line 69: The first sentence of the new paragraph should be included as the last sentence of the previous paragraph. Begin the new paragraph with the sentence ‘’This study examines ………………. ‘’

• Line 89-‘’along with other groups’’: Mention the other groups.

• Line 147: Replace ‘’Based on the aforementioned,’’ with ‘’Therefore,’’.

• Methods-Line 162: Who exactly is a provider here?

• Line 168: Replace ‘’12.000’’ with ‘’12,000’’ if you are referring to twelve thousand.

• Lines 168-170: In addition, would you prefer to mention the related number at the end of 2020?

• Line 175: On what basis 1,500 panel members were selected/considered out of the total pool of 12,000 participants (panel members)?

• Lines 177-179: Weren’t the completed questionnaires anonymous in nature?

• Line 187: Three items were negative statements and seven items were positive statements. In other words, most of the statements were positive statements. What impact does that have on your findings? It can affect what is termed 'acquiescence bias'. Such surveys/questionnaires have a more balanced number of negative and positive statements.

• Line 195: Use the journal style while citing a reference.

• Results-Line 223: In addition, provide the mean(SD) age separately for male and female.

• Table 3: The last column consists of % and mean(SD). Indicate ‘’mean(SD)” with * both in the heading of the column and for the related values.

• Table 3: With reference to the rows, provide the row related to ‘’trust’’ before ‘’involvement in medical-decision making’’.

• Table 4: Replace ‘’Men’’ with ‘’Male’’ for uniformity’s sake when compared to Table 3 and other Tables.

• Besides, address the reviewers’ comments.

• In addition, let us know the etiology for the delay in submission of this manuscript (data collection: beginning of 2016; manuscript submission: beginning of 2021).

Reviewers' comments:

Reviewer's Responses to Questions

**Comments to the Author**

1. Is the manuscript technically sound, and do the data support the conclusions?

Reviewer #1: Yes

Reviewer #2: Yes

2. Has the statistical analysis been performed appropriately and rigorously? 

Reviewer #1: Yes

Reviewer #2: I Don't Know

3. Have the authors made all data underlying the findings in their manuscript fully available?

Reviewer #1: No

Reviewer #2: Yes

4. Is the manuscript presented in an intelligible fashion and written in standard English?

Reviewer #1: Yes

Reviewer #2: Yes

5. Review Comments to the Author

Reviewer #1: Overall, this is a well-written manuscript that confirms earlier findings on the relationship between trust and shared-decision making. The findings, such as the relationship between trust, SDM, and gender, are important contributions towards understanding the factors/determinants of SDM. I have no further comments on the content and recommend this manuscript for publication.

With regard to the availability of data, the authors should provide an ethical/legal justification for not making it accessible.

Reviewer #2: Overall, this is a very interesting study and will be a valuable addition to the literature on SDM and MDM. I congratulate the authors.

I have one query:

Were the survey participants asked to respond about a particular physician, e.g. their General Practitioner? If no, would the survey participants likely have multiple physicians? (In some countries patients have a General Practitioner and specialist physicians treating them at the same time).

And one comment:

It would be interesting to conduct a qualitative study to understand why women with high trust are more involved in MDM compared to men with high trust – this may help with strategies to reverse men’s less than optimal involvement in MDM that you mention in the last paragraph of page 16 of the PDF.

6. PLOS authors have the option to publish the peer review history of their article (what does this mean?). If published, this will include your full peer review and any attached files.

Reviewer #1: No

Reviewer #2: No

---

## [Author Response · Author response to Decision Letter 0]

10 Aug 2021

Dear Dr. Chenette,

We would like to re-submit the revised manuscript entitled “Relationship between Trust and Patient Involvement in Medical Decision-Making: A Cross-Sectional Study” drafted by I. Pokhilenko, Dr. T.E.M. van Esch, Dr. A.E.M. Brabers, and Prof. J.D. de Jong for publication as a research article in PLOS ONE. We are happy to have received the positive feedback and the suggestions for further improvements. All comments were used to revise the paper. In the response letter below, we provided a response to each comment and described how we revised the paper accordingly. In the main file, the revised sections are marked in track changes. 

Some comments referred to the availability of the data used in this study. The data for this study was collected in the Dutch Heath Care Consumer Panel of the Netherlands Institute for Health Services Research (Nivel). We will upload the minimal anonymized data set to replicate our study findings to the DANS EASY repository in the upcoming weeks. This dataset will be made available upon request. Additionally, the same minimal anonymized data set is directly available upon request from prof. Judith D. de Jong (j.dejong@nivel.nl), project leader of the Dutch Health Care Consumer Panel, or the secretary of this panel (conusmentenpanel@nivel.nl). The Dutch Health Care Panel had a program committee, which supervises processing the data of the Dutch Health Care Consumer Panel and decides about the use of the data. This program committee consists of representatives of the Dutch Ministry of Health, Welfare and Sport, the Health Care Inspectorate, Zorgverzekeraars Nederland (Association of Health Care Insurers in the Netherlands), the National Health Care Institute, the Federation of Patients and Consumer Organisations in the Netherlands, the Dutch Healthcare Authority and the Dutch Consumers Association. All research conducted within the Consumer Panel has to be approved by this program committee. The committee assesses whether a specific research fits within the aim of the Consumer Panel, which is to strengthen the position of the health care user.

We also revised the reference list. Reference #44 has been updated from “CCMO. Your research: does it fall under the WMO? [cited 2018 May 8]. Available from: http://www.ccmo.nl/en/your-research-does-it-fall-under-the-wmo” to “CCMO. Your research: Is it subject to the WMO or not? [cited 2021 July 6]. Available from: https://english.ccmo.nl/investigators/legal-framework-for-medical-scientific-research/your-research-is-it-subject-to-the-wmo-or-not” as the original link was no longer available. Furthermore, we believe to have followed the PLOS ONE’s style requirements. 

With the aim of reaching the broad multidisciplinary audience of your journal, we would be grateful if you would consider this submission for publication in PLOS ONE. Hereby, we confirm that each author contributed to the conception, design, analysis and interpretation of data and the writing of the paper. All authors have approved the submitted version of the manuscript. The manuscript is an original contribution and is not being considered for publication elsewhere.

I look forward to hearing from you, on behalf of the co-authors,

Yours sincerely, 

Irina Pokhilenko

 

Journal Requirements:

 

Additional Academic Editor Comments:

• Abstract-Line 33: Mention ‘’Flynn et al. (2006).’’ as a part of a complete sentence (avoid the bracket and accordingly edit the sentence.

Author response: We thank the editor for this and other comments. The sentence was revised.

“Patient involvement was measured using two items based on the study published by Flynn and colleagues in 2006.” (lines 33-34)

• Introduction-Line 69: The first sentence of the new paragraph should be included as the last sentence of the previous paragraph. Begin the new paragraph with the sentence ‘’This study examines ………………. ‘’

Author response: We revised the paragraphs accordingly. 

• Line 89-‘’along with other groups’’: Mention the other groups.

Author response: The sentence was revised accordingly.

“Furthermore, previous studies have been performed either among specific patient groups including patients of breast cancer, prostate cancer and fracture outpatient clinics, African-American patients with diabetes along with other groups, patients with cardiovascular disease or among a group of breast cancer specialists.” (lines 88-91)

• Line 147: Replace ‘’Based on the aforementioned,’’ with ‘’Therefore,’’.

Author response: The sentence was revised accordingly. 

“Therefore, we hypothesize that: …” (line 150)

• Methods-Line 162: Who exactly is a provider here?

Author response: The correct term would be “address supplier”. We revised the sentence accordingly. 

“First, addresses of potential participants are bought from an address supplier.” (lines 164-165)

• Line 168: Replace ‘’12.000’’ with ‘’12,000’’ if you are referring to twelve thousand.

Author response: The sentence was revised accordingly.

“At the moment of data collection (February 2016), the Dutch Health Care Consumer Panel consisted of approximately 12,000 members of 18 years or older.” (lines 170-172)

• Lines 168-170: In addition, would you prefer to mention the related number at the end of 2020?

Author response: The number of participants as of 2021 is approximately 11,500. We mentioned this in the text. 

 “As of 2021, the number of panel members is 12,000.” (line 172)

• Line 175: On what basis 1,500 panel members were selected/considered out of the total pool of 12,000 participants (panel members)?

Author response: The sample was selected randomly. The only criteria was to ensure that the group was representative of the Dutch adult population. We added a sentence to clarify this. 

“The sample was randomly selected and was representative of the Dutch adult population for age and sex.” (lines 180-181)

• Lines 177-179: Weren’t the completed questionnaires anonymous in nature?

Author response: The questionnaires were anonymous. Besides, the researchers who analysed the data did not have access to the identifiable data of the respondents. 

“The questionnaires used in this study were anonymous in nature. Besides, the researchers who analyse the data do not have access to any identifiable information of the participants (e.g. name and address).” (lines 177-179)

• Line 187: Three items were negative statements and seven items were positive statements. In other words, most of the statements were positive statements. What impact does that have on your findings? It can affect what is termed 'acquiescence bias'. Such surveys/questionnaires have a more balanced number of negative and positive statements.

Author response: We reflected on this aspect in the discussion section.

“It is important to note that WF-D was not fully balanced in terms of the nature of the statements. It contained seven positive statements and three negative statements, which might have led to acquiescence bias. However, WF-D has been validated [46] and frequently used for trust research [49, 50].” (lines 334-337)

• Line 195: Use the journal style while citing a reference.

 Author response: The reference was revised.

“Two questions based on the items developed by Flynn et al [9] were used to measure patient involvement in MDM (Table 2).” (line 201)

• Results-Line 223: In addition, provide the mean(SD) age separately for male and female.

Author response: We added two rows in Table 3 with the mean age and SD separately for male and female respondents.

• Table 3: The last column consists of % and mean(SD). Indicate ‘’mean(SD)” with * both in the heading of the column and for the related values.

Author response: Table 3 was revised accordingly.

• Table 3: With reference to the rows, provide the row related to ‘’trust’’ before ‘’involvement in medical-decision making’’.

 Author response: Table 3 was revised accordingly.

• Table 4: Replace ‘’Men’’ with ‘’Male’’ for uniformity’s sake when compared to Table 3 and other Tables.

Author response: Table 4 was revised accordingly.

• Besides, address the reviewers’ comments.

Author response: We believe to have addressed all reviewers’ comments and explained how we addressed them in the note below.

• In addition, let us know the etiology for the delay in submission of this manuscript (data collection: beginning of 2016; manuscript submission: beginning of 2021).

Author response: The data used in this study was originally collected in 2016. The current study was conducted based on secondary data analysis in 2018. The delay in manuscript submission occurred due to the delays in the internal review process. However, we believe that this delay does not have impact on the relevance of the results.

 

Reviewers' comments:

Reviewer's Responses to Questions

Comments to the Author

1. Is the manuscript technically sound, and do the data support the conclusions?

Reviewer #1: Yes

Reviewer #2: Yes

2. Has the statistical analysis been performed appropriately and rigorously? 

Reviewer #1: Yes

Reviewer #2: I Don't Know

3. Have the authors made all data underlying the findings in their manuscript fully available?

Reviewer #1: No

Reviewer #2: Yes

4. Is the manuscript presented in an intelligible fashion and written in standard English?

Reviewer #1: Yes

Reviewer #2: Yes

5. Review Comments to the Author

Reviewer #1: Overall, this is a well-written manuscript that confirms earlier findings on the relationship between trust and shared-decision making. The findings, such as the relationship between trust, SDM, and gender, are important contributions towards understanding the factors/determinants of SDM. I have no further comments on the content and recommend this manuscript for publication.

With regard to the availability of data, the authors should provide an ethical/legal justification for not making it accessible.

Author response: The authors thank the reviewer for taking time to review the manuscript and for recommending it for publication. 

With regard to the availability of data, we will upload the minimal anonymized data set to replicate our study findings to the DANS EASY repository. This dataset will be made available upon request. Additionally, the same minimal anonymized data set is directly available upon request from prof. Judith D. de Jong (j.dejong@nivel.nl), project leader of the Dutch Health Care Consumer Panel, or the secretary of this panel (conusmentenpanel@nivel.nl). The Dutch Health Care Panel had a program committee, which supervises processing the data of the Dutch Health Care Consumer Panel and decides about the use of the data. This program committee consists of representatives of the Dutch Ministry of Health, Welfare and Sport, the Health Care Inspectorate, Zorgverzekeraars Nederland (Association of Health Care Insurers in the Netherlands), the National Health Care Institute, the Federation of Patients and Consumer Organisations in the Netherlands, the Dutch Healthcare Authority and the Dutch Consumers Association. All research conducted within the Consumer Panel has to be approved by this program committee. The committee assesses whether a specific research fits within the aim of the Consumer Panel, which is to strengthen the position of the health care user.

Reviewer #2: Overall, this is a very interesting study and will be a valuable addition to the literature on SDM and MDM. I congratulate the authors.

I have one query:

Were the survey participants asked to respond about a particular physician, e.g. their General Practitioner? If no, would the survey participants likely have multiple physicians? (In some countries patients have a General Practitioner and specialist physicians treating them at the same time).

Author response: The authors thank the reviewer for taking time to review the manuscript and for recommending it for publication. This study was conducted among the general Dutch population. The questionnaire in Dutch did not refer to any particular physician. In the Dutch healthcare system, every person is only able to be assigned to one general practitioner at a time. Therefore, we assume that the respondents provided responses regarding their general practitioner. However, we agree that in case one is treated by multiple physicians, there might be some confusion. We reflected on this point in the discussion.

“Both instruments used in this study did not refer to a particular physician (e.g. respondent’s general practitioner). This might have affected the accuracy of the responses in case some of the respondents had more than one treating physician.” (lines 347-349)

And one comment:

It would be interesting to conduct a qualitative study to understand why women with high trust are more involved in MDM compared to men with high trust – this may help with strategies to reverse men’s less than optimal involvement in MDM that you mention in the last paragraph of page 16 of the PDF.

Author response: We thank reviewer for this comment. We agree that such study could be very interesting. We added this point in the discussion.

“Employing a qualitative study design could also be interesting for investigating the reasons for why women with high trust in their physician are more involved in MDM compared to men with high trust. This could help design strategies for facilitating the involvement of men with high trust in MDM.” (lines 360-363)

6. PLOS authors have the option to publish the peer review history of their article (what does this mean?). If published, this will include your full peer review and any attached files.

Do you want your identity to be public for this peer review? For information about this choice, including consent withdrawal, please see our Privacy Policy.

Reviewer #1: No

Reviewer #2: No

---

## [Editor Report · Decision Letter 1]

13 Aug 2021

Relationship between Trust and Patient Involvement in Medical Decision-Making: A Cross-Sectional Study

PONE-D-21-10608R1

Dear Dr. Pokhilenko,

We’re pleased to inform you that your manuscript has been judged scientifically suitable for publication and will be formally accepted for publication once it meets all outstanding technical requirements.

Kind regards,

Prof. Ritesh G. Menezes, M.B.B.S., M.D., Diplomate N.B.

Academic Editor

PLOS ONE

---

## [Editor Report · Acceptance letter]

17 Aug 2021

PONE-D-21-10608R1 

Relationship between trust and patient involvement in medical decision-making: A cross-sectional study 

Dear Dr. Pokhilenko:

I'm pleased to inform you that your manuscript has been deemed suitable for publication in PLOS ONE. Congratulations! Your manuscript is now with our production department. 

Kind regards, 

on behalf of

Prof. Dr. Ritesh G. Menezes 

Academic Editor

PLOS ONE